# An Approach to Improving GNSS Positioning Accuracy Using Several GNSS Devices

**María Jesús Jiménez-Martínez** [1,*] **, Mercedes Farjas-Abadia** [2] **and Nieves Quesada-Olmo** [1]

1 Department of Cartographic Engineering, Geodesy and Photogrammetry, Universitat Politècnica de València, 46005 València, Spain; niequeol@cgf.upv.es
2 Department of Surveying and Cartographic Engineering, Universidad Politécnica de Madrid, 28031 Madrid, Spain; m.farjas@upm.es
* Correspondence: mjjimenez@cgf.upv.es; Tel.: +34-6447-716-877

**Abstract:** Single point positioning (SPP) mode, related to pseudorange measurements, limits the level of accuracy to several meters in open sky and to several dozens of meters in urban canyons. This paper explores the effect of using a large number of SPP observations from low-cost global navigation system (GNSS) receivers, smartphones, and handheld GNSS units. Data segmentation and bootstrapping statistical methods were used to obtain the deviation, which can describe the accuracy of the large sample. The empirical test recording data showed that the error may achieve a sub-meter horizontal accuracy by the simple process of increasing the measurements of smartphones and handheld GNSS units. However, the drawback is the long period of time required. To reduce the satellite tracking time, a least squares solution network was applied over all the recorded data, assisted by the external geometric conditions. The final goal was to obtain the absolute positioning and associated deviations of one vertex from three or five GNSS receivers positioned on a network. The process was tested in three geodetic network examples. The results indicated that the enhanced SPP mode was able to improve its accuracy. Errors of several meters were reduced to values close to 50 cm in 25–37 min periods.

**Keywords:** GNSS; single point positioning; bootstrapping; smartphone; mobile technology; least squared; geometric constraints

## 1. Introduction

Global navigation satellite systems (GNSS) are one of the most pervasive technologies in use. Applications of satellite navigation are integrated in billions of smartphones and other devices. GNSS have become an integral part of everyday life helping us in very different areas, including transportation, precise machine control, and surveying. According to Carlo des Dorides, Executive Director of the European GNSS Agency (GSA), for the year 2020 the global installed base of GNSS devices in use was forecast to reach almost 6.5 billion [1].

To find the absolute position of a point is a very fundamental problem in positional GNSS. The most common GNSS absolute positioning is called single point positioning (SPP). In fact, SPP mode is usually adopted by mass-market receivers. SPP mode is only related to pseudorange measurements. This mode limits their level of accuracy to several meters in open sky and to several dozens of meters in urban canyons [2]. The pseudorange noise of smart devices is about 10 times larger than that of geodetic receivers [3].

Currently, smartphones have real operational capability of more GNSS, which will enhance their accuracy, especially for situations with bad satellite visibility and/or multipath errors [4,5]. Thus, the horizontal position errors can decrease 35%, for single frequency receivers, and 40% for dual frequency receivers [6,7]. This implies an advantage of these inexpensive receivers.

Another GNSS absolute positioning mode is the precise point positioning technique (PPP). Different researchers are exploring the application of PPP in dual-frequency smartphones [8–13]. Carrier phase and pseudorange observations are needed to achieve centimeter-level accuracy. The PPP technique also uses high-precision IGS (International GNSS Service) products with a single receiver. Wu et al. [9] evaluated the PPP solution in post-processing, using a dual frequency smartphone in a static position, and could perform a horizontal accuracy of 0.50 m in a 107 min convergence period. Psychas et al. [14] required only 34 min to achieve a positioning error below 1 m.

Several research works obtained cm-accuracy with smartphones. Darugna et al. [15] concluded that centimeter RTK-PPP level positioning accuracies can be achieved with smartphones if the level of multipath is sufficiently reduced. Wanninger and Heßelbarth [16] were able to demonstrate centimeter-accurate position determination with code and carrier phase observations from a Huawei P30, but it can only be achieved if the ambiguities are fixed to their correct integer values [17]. Unfortunately, not all carrier phase observations have the property of integer ambiguities. The frequent signal interruptions and carrier phase cycle slips do not always help to solve carrier phase ambiguity [8–12]. Finally, only a few mobile phones can write phase observations including the Xiaomi MI8. We would point out that not requiring carrier phase observations by SPP mode could represent an advantage.

Additionally, it is expected that fifth-generation (5G) positioning will be able to provide more precise location and enhanced availability in all kinds of environment, especially urban canyons where the base stations are close enough and most GNSS signals are blocked and suffer from severe multipath conditions. In this frame, the design made during the 5G-Champion project [18] has led to a solution combining both positioning systems. The 5G observations provide extra equations added to both SPP and PPP solutions [19]. The addition of a 5G antenna clearly improves the positioning accuracy in the order of 1 m or even below [19]. In the connected vehicle scenario, a hybrid positioning 5G-GNSS is presented in [20], and results indicate that beyond a certain value of the carrier-to-noise signal ratio (CNR), a practical 5G positioning can improve its accuracy through GNSS integration. Improving the accuracy of SPP mode could improve positioning based on the hybridization of 5G and GNSS observations.

The key issue in this paper is how to use a low-cost receiver in the operational mode SPP and achieve a sub-meter horizontal accuracy, with the aim of obtaining absolute positioning as the PPP technique does. The advantages of working under SPP include not requiring carrier phase observations, external corrections or high-precision products, especially when the internet is not available. To this end, we proposed a least squares method. The method is based on a long static SPP observation series from several GNSS receivers and the integration of geometric constraints. The constraints are the distances among receivers.

The first question is if a large quantity of GNSS data will improve the randomness of the sample and the accuracy of the measurements' mean. In accordance with Mayer-Schönberger and Cukier [21] we will have less sampling errors using all the data, instead of only a small portion. With less error from population sampling, we can accept more error from the measurements, and the average of the values can be more accurate. In this paper, we explore the effect of using a large amount of SPP observations from low-cost GNSS receivers, smartphones, and handheld GNSS units. Long static data were collected, and we performed statistical analyses with the following aims: to evaluate the accuracy, to avoid a false value of the variance from a large amount of data, and to check these results using bootstrapping. However, the drawback is the long period of time required.

To reduce the satellite tracking time, a least squares solution network was applied over all the recorded data, and assisted by external geometric conditions. The final goal was to obtain the position and associated standard deviation of one vertex from receivers positioned on a network.

The paper is structured as follows. Section 2.1 provides the materials and methods necessary to obtain the mean and variance from a large amount of SPP observations. All measurements were taken in a static position. Section 2.2 provides with the procedure to introduce constraints in a geodetic network from several GNSS receiver devices, to increase the amount of data in less time. In Section 3, we evaluate the results, applying the theory presented above. Finally, in Section 4 and, Section 5, respectively, the discussion is drawn, and our conclusions are presented.

## 2. Materials and Methods

### 2.1. Data Segmentation, Variance Value and Bootstrapping

Accurate assessment of estimate quality has been a longstanding concern in statistics. In this section, we describe two processes to obtain the variance from a large amount of data: variance value and the bootstrapping technique.

### 2.1.1. Data Segmentation and Variance Value

The mean from large observation data must be estimated to evaluate the quality. As is well known the calculation of variance is the average distance observed where the values $x$ are from a central point $\overline{x}$ (compare with Equation (1)):

$$variance = \sigma^2 = \frac{\sum(x - \overline{x})^2}{n - 1} \tag{1}$$

where $x =$ the observed values, $\overline{x} =$ the mean of the observed values $n =$ the number of observations of the sample, and $\sigma =$ standard deviation.

Due to the sample size $n$ in the denominator, big data often produces extremely small variances. These highly significant results caused by near zero variances give the researcher a false sense of precision [22].

McFarland and McFarland [22] proposed the data segmentation of populations, and this is a well-known problem in statistics when analyzing observational data in an effort to make it more representative of the population. While these suggestions will not remove all the errors found in the analysis of large amounts of data, they will improve the quality of the statistic results identifying and addressing bias. In our study, the segmentation of populations will be in terms of the time and, consequently, the number of observations. Theoretically, as the amount of data increases, the quality of the coordinate means will improve.

Thus, the first step is the process of dividing up the data and grouping similar data together. Afterward, each group or segment is considered as one observation, allowing us to determine the variance of the sample using Equation (1). In the next section, an alternative way to obtain the variance from the whole segments and compare the results is proposed.

### 2.1.2. Data Segmentation and Bootstrapping

Once the segment number and size are determined, the bootstrapping method is recommended to obtain the variance of the sample [23]. The bootstrap method is perhaps the best known and most widely used among classical methods, due to its simplicity, generic applicability, and automatic nature [24]

The core idea of the bootstrap technique is to make certain kinds of statistical inference with the help of modern computer power. The method was introduced by Efron in 1979, originally conceived to empirically estimate the variance of an estimator from a number of samples drawn with replacements from the same data. Since then, thanks to its simple implementation and strong performance, the bootstrap idea has been widely adopted and inflected in many variations, to the point that now bootstrapping broadly refers to a large number of estimation strategies that use resampling for a wide range of applications, including land cover classification accuracy from remote sensing [25] strategies to reduce the effect of large blunders in GNSS absolute positioning [26], methods to estimate absolute positions using GNSS and Inertial Navigation Systems [27], and methods for GNSS carrier-

phase ambiguity resolution [28,29], among others, providing considerable potential in geographic information science.

The basic idea of bootstrapping in our research involved the following steps:

1. Begin with a sample *S* from a population with *n* GNSS observations, thus, the sample size is *n*.
2. Draw a sample from the original sample *S* data with replacement size *n*, and replicate *k* times. Each re-sampled is called a Bootstrap Sample, and there will a total of *k* bootstrap samples.
3. Evaluate the mean for each bootstrap sample; there will be a total *k* means. A few thousands is considered a reasonable value for *k* to have a good approximation [22].
4. Estimate the variance of the *k* means. With the aid of a computer, we can make *k* as large as we like to approximate to the sampling distribution of our estimator mean. Our goal is to estimate the variance of the mean sample generated and compare it with the variance obtained from Equation (1).

### 2.2. Sequential Solution Method in a Local Geodetic Network. Mathematical and Stochastic Models

Mobile mapping systems have increased the accuracy of direct georeferencing with geometric scene constraints [30]. Smartphone applications for bridge damage detection have been used with position conditions [31]. The analysis of GNSS positioning can use a baseline length constraint [32], an affine-constrained GNSS attitude model [33], or a GNSS monitoring system to control high-rise buildings [34]. Therefore, constraints have a considerable potential to improve the accuracy of different types of sensor.

GNSS sensors of smartphones and other devices suffer from poor accuracy, accumulated drift, and high signal noise. The Big Data theory says that a superior quantity of data provided by a greater number of sensors will improve the accuracy of the mean measurements. To increase the number of sensors, we proposed a GNSS network to determine the positions. The least squares method together with geometric constraints may give the adjusted coordinates all the vertices of the defined network. The geometric constraints are linked to the spatial positions.

In summary, the solution obtained will come from the mathematical model of the initial adjustment, in addition to a second group of constraint functions that make up the model of sequential solution methods [35]. Both groups have a common set of parameters.

The constraints will be conditions that can be described by an equation that must absolutely, positively be true no matter what our solution is, and thus the results will be more robust. Therefore, the first step is to identify the constraints, which are the quantities that must be true regardless of the solution. The known shape of the geodetic network defines these specific conditions (see Figure 1), which are the distances between GNSS receivers. Distances are measured with accuracy sufficient to treat the quantities as relatively error-free, compared to the GNSS observations in the network [36].

Least squares adjustment deals with two equally important components: the stochastic model, the mathematical model and the constraint conditions. The stochastic model introduces information regarding the precision of the observations. The mathematical model expresses the relations between the observations and parameters (unknowns), such as the coordinates, distances, angles, and heights.

We first specify the observation equation model. The mixed model and the sequential mixed model are the base models from which other solutions can be conveniently derived by appropriate specifications [35,37]. Assuming that observations are made in two groups $l_{1a}$ and $l_{2a}$, then two mixed adjustment models can be written as:

$$
\begin{aligned}
f_1(l_{1a}, x) &= 0 \\
f_2(l_{2a}, x) &= 0
\end{aligned}
\tag{2}
$$

Both groups have a common set of parameters *x*.

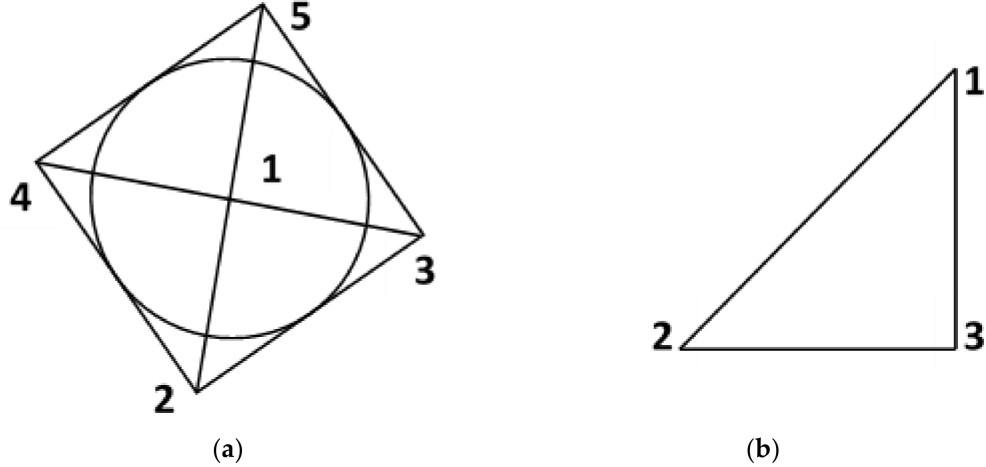

**Figure 1.** From left to right, two examples of global navigation satellite systems (GNSS) networks: (**a**) a five vertices network and (**b**) a three vertices network.

The a priori variance of the unit weight should be the same for both groups, and so the matrix weight is:

$$P = \begin{pmatrix} P_1 & 0 \\ 0 & P_2 \end{pmatrix} = \tag{3}$$

The linearization of Equation (2) yields:

$$B_1 \cdot v_1 + A_1 \cdot x + w_1 = 0 \tag{4}$$

$$B_2 \cdot v_2 + A_2 \cdot x + w_2 = 0 \tag{5}$$

where:

$$B_1 = \left. \frac{\partial f_1}{\partial l_1} \right|_{l_{1b}, x_0} \quad A_1 = \left. \frac{\partial f_1}{\partial x} \right|_{l_{1b}, x_0} \quad w_1 = f_1(l_{1b}, x_0) \tag{6}$$

$$B_2 = \left. \frac{\partial f_2}{\partial l_2} \right|_{l_{2b}, x_0} \quad A_2 = \left. \frac{\partial f_2}{\partial x} \right|_{l_{2b}, x_0} \quad w_2 = f_2(l_{2b}, x_0) \tag{7}$$

The solution is obtained by introducing vectors of the Lagrange multipliers, $\lambda_1$ and $\lambda_2$, and minimizing the function:

$$\phi(v_1, v_2, \lambda_1, \lambda_2) = v_1^T \cdot v_1 + v_2^T \cdot v_2 - 2\lambda_1^T \cdot (B_1 \cdot v_1 + A_1 \cdot x + w_1) - 2\lambda_2^T \cdot (B_2 \cdot v_2 + A_2 \cdot x + w_2) \tag{8}$$

The partial derivatives of Equation (8) to zero are:

$$\frac{1}{2} \cdot \frac{\partial \phi_1}{\partial v_1} = v_1 - B_1^T \cdot \lambda_1 = 0 \tag{9}$$

$$\frac{1}{2} \cdot \frac{\partial \phi_1}{\partial v_2} = v_2 - B_2^T \cdot \lambda_2 = 0 \tag{10}$$

$$\frac{1}{2} \cdot \frac{\partial \phi_1}{\partial x} = -A_2^T \cdot \lambda_1 - A_2^T \cdot \lambda_2 = 0 \tag{11}$$

$$\frac{1}{2} \cdot \frac{\partial \phi_1}{\partial \lambda_1} = B_1^T \cdot v_1 - A_1 \cdot x + w_1 = 0 \tag{12}$$

$$\frac{1}{2} \cdot \frac{\partial \phi_1}{\partial \lambda_2} = B_2^T \cdot v_2 - A_2 \cdot x + w_2 = 0 \tag{13}$$

solving for $v_1$, $v_2$, $\lambda_1$, $\lambda_2$ and $x$. Equations (9) and (10) give the residuals

$$v_1 = P_1^{-1} \cdot B_1^T \cdot \lambda_1 \tag{14}$$

$$v_2 = P_2^{-1} \cdot B_2^T \cdot \lambda_2 \tag{15}$$

Combining Equations (14) and (12) yields

$$M_1 \cdot \lambda_1 + A_1 \cdot \hat{x} + w_1 = 0 \tag{16}$$

where

$$M_1 = B_1 \cdot P_1^{-1} \cdot B_1^T \tag{17}$$

is an $r_1$x $r_1$ symmetric matrix. The Lagrange multiplier becomes:

$$\lambda_1 = -M_1 \cdot A_1 \cdot x - M_1^{-1} \cdot w_1 \tag{18}$$

Equations (11) and (13) become, after combination with Equations (18) and (15):

$$A_1^T \cdot M_1^{-1} \cdot A_1 \cdot x + A_1^T \cdot M_1^{-1} \cdot w_1 - A_2^T \cdot \lambda_2 = 0 \tag{19}$$

$$B_2 \cdot P_2^{-1} \cdot B_2^T \cdot \lambda_2 + A_2 \cdot x + w_2 = 0 \tag{20}$$

By using

$$M_2 = B_2 \cdot P_2^{-1} \cdot B_2^T \tag{21}$$

we can write Equations (19) and (20) in matrix form:

$$\begin{bmatrix} A_1^T \cdot M_1^{-1} \cdot A_1 & A_2^T \\ A_2 & -M_2 \end{bmatrix} \cdot \begin{bmatrix} x \\ \lambda_2 \end{bmatrix} = \begin{bmatrix} A_1^T \cdot M_1^{-1} \cdot w_1 \\ w_2 \end{bmatrix} \tag{22}$$

Equation (22) shows how the normal matrix of the first group must be augmented to find the solution of both groups. Equation (22) can be used to incorporate exterior information regarding the vector of parameters *x*.

In our particular case, the observation equation model with conditions (compare with Equations (4) and (5)) is:

$$A_1 \cdot x - v_1 - L_1 = 0 \tag{23}$$

$$A_2 \cdot x \quad - L_2 = 0 \tag{24}$$

where $A_1$ = the coefficient matrix, $x$ = vector of the unknowns as coordinates of network vertices, $v_1$ = vector of the residuals , $L_1$ = vector of the independent terms as observed coordinates , $A_2$ = the matrix used to incorporate exterior information about variables relations, and $L_2$ = vector of the independent terms. It is customary to denote the discrepancy by $L$, instead of w, when dealing with the observation equation model.

Taking the definition of the matrix $B_1$ (compare with Equation (6)), $B_1$ will be the identity matrix: $B_1 = I$ .

$A_2$ indicates that the solution of $x$ depends on the special condition implied by the $A_2$ matrix. The only specifications for implementing the conditions are $P_2^{-1}$. Therefore, according to Equations (17) and (21):

$$M_1 = P_1^{-1} \tag{25}$$

$$M_2 = 0 \tag{26}$$

From Equation (22) the equation model with conditions is now:

$$\begin{bmatrix} A_1^T \cdot P_1 \cdot A_1 & A_2^T \\ A_2 & 0 \end{bmatrix} \cdot \begin{bmatrix} x \\ \lambda_2 \end{bmatrix} = \begin{bmatrix} A_1^T \cdot P_1 \cdot L_1 \\ L_2 \end{bmatrix} \tag{27}$$

The one-step solution is given by Equation (27). Now, we can obtain the adjusted coordinates of the *n* vertices of the network.

The equation model $f(x) = l$ is defined by the coefficient matrix $A_1$:

$$f(x) \equiv X_i - (X_{io} + v_i) = 0 \tag{28}$$

The coefficient matrix $A_2$ establishes the conditions of the mathematical model $g(x) = 0$:

$$g(x) \equiv X_{io} - X_{jo} - \Delta X_{ij} = 0 \tag{29}$$

where $X_i$ = the calculated coordinates of the $i$ vertex, $X_{io}$ = the observed coordinates of the $i$ vertex, $v_i$ = the residual of $X_{io}$, and $i \in 1, 2, 3 \dots n$-vertex. $X_{jo}$ = the observed coordinates of the $j$ vertex, $\Delta X_{ij}$ = a known and constant value, and $i \in 1, 2, 3 \dots n$-vertex.

From the vertex coordinates and known distances between vertices, the coordinate increments $\Delta X_{ij}$ can be calculated, which will be described in Section 3.4.

$f(x)$ and $g(x)$ are linear mathematical models.

The stochastic model introduces information regarding the precision of the observations. As for the precision of the network results we are using the expression of the variance–covariance matrix $\Sigma_x$. In our particular case of the observation equation model with conditions, the cofactor matrix $Q_x$ is:

$$Q_x = Q_{x*} + \Delta Q_x = I + \left( I \cdot A_2^T \cdot T^{-1} \cdot A_2 \cdot I \right) \tag{30}$$

where,

$$Q_{x*} = A_1^T \cdot P_1^{-1} \cdot A_1 = N_1^{-1} \tag{31}$$

$$T = A_2 \cdot N_1^{-1} \cdot A_2{}^T \tag{32}$$

$$\Delta Q_x = -N_1^{-1} \cdot A_2^T \cdot T^{-1} \cdot A_2 \cdot N_1^{-1} \tag{33}$$

The a posteriori variance of unit weight is computed in the usual way:

$$\sigma_0^2 = \frac{v_1^T \cdot P_1 \cdot v_1}{n_1 + n_2 - u} \tag{34}$$

where $v_1$ is the vector of residuals from $f(x)$ model, $n_1$ and $n_2$ are the number of equations, and $u$ denotes the number of parameters.

The stochastic model of the observations is the expression of the variance–covariance matrix, as usual:

$$\Sigma_x = \sigma_0^2 \cdot Q_x \tag{35}$$

## 3. Results

This section contains a summary of the results obtained applying the above presented theory.

### 3.1. Large Amounts of Data in Stationary Positions

To evaluate the accuracy, an empirical test recording large amounts of data from different GNSS devices, in stationary positions was developed. The key questions are that if big data is used for the mean, which will define the position, would it produce better accuracy? Could this produce sub-meter accuracy and how long it is necessary to achieve this improvement?

For comparison studies, five different devices were used. Two low-cost handheld GNSS units: Garmin Etrex 30 and Trimble Geo XT, and three smartphones: Samsung Galaxy S3, Huawei Y330-U01, and Xiaomi Mi 8 dual frequency, all of them listed in Table 1.

**Table 1.** The handheld global navigation satellite systems (GNSS) and smartphones used in the empirical tests. The tolerance of each GNSS device. The tolerance value will be the quadratic composition of the devices' dimension.

| GNSS Device | First Release Year | Tolerance (m) |
| --- | --- | --- |
| Garmin Etrex 30 (0.103 m × 0.054 m) | 2015 | $\sqrt{0.103^2 + 0.054^2} = 0.1163$ |
| Trimble Geo XT (0.255 m × 0.127 m) | 2005 | $\sqrt{0.255^2 + 0.127^2} = 0.2849$ |
| Samsung Galaxy S3 (0.137 m × 0.071 m) | 2011 | $\sqrt{0.137^2 + 0.071^2} = 0.1543$ |
| Huawei Y330-U01 (0.122 m × 0.064 m) | 2014 | $\sqrt{0.122^2 + 0.064^2} = 0.1378$ |
| Xiaomi Mi 8 (0.155 m × 0.075 m) | 2018 | $\sqrt{0.155^2 + 0.075^2} = 0.1722$ |

The Xiaomi Mi 8 can receive L1/E1/and L5/E5 signals from GPS, Galileo, BeiDou, and GLONASS satellites. Previous studies [6,7,38] demonstrated that new smartphones equipped with multi-constellation dual frequency GNSS reached a positioning accuracy around 1.5–10 m, as the horizontal root mean square error in SPP mode. Psychas et al. [18] achieved errors from 1.13 to 1.85 m, in open sky environments, with a clear superiority of the L5/E5 signals.

Garmin Etrex 30, Trimble Geo XT, Samsung Galaxy S3 and Huawei Y330-U01 can receive only L1 signals from GPS and Glonass. Garmin Etrex 30 can also receive European Geostationary Navigation Overlay Service (EGNOS) data.

The smartphones and GPS units were positioned on the rooftop of a four-story building, with clear visibility into the sky.

GNSS data was collected using our own app that logs the time, latitude, and longitude. The recorded data were converted into the universal transverse Mercator (UTM) projection. We have projected the WGS84 geographic coordinates into UTM using the equations of the representation (EPSG projection 32630).

For reference data, a professional grade Trimble R10 device was used with real-time GNSS techniques. The coordinates of the ground control points obtained had a short-term accuracy of 0.002–0.03 m. However, the location of the GNSS antennas within the five low-cost devices is not known. The true coordinates of the five low-cost devices are unknown, as the exact position of the antenna phase centers are unknown, while also the devices were located next to and not exactly on the coordinate-known vertices. The assumption made is to consider the tolerance for the precision of positioning as the size of each GNSS device. The tolerance value will be the quadratic composition of the devices' dimension (compare with Table 1). All tolerance values were below 0.50 m.

Figures 2 and 3 shows the relationship between the time and the quadratic composition of the East and North errors. The East and North errors are the differences between the control point and coordinate means. All devices taking a measurement every 10 s, but the Xiaomi Mi 8 logging rate was set to 1 s.

The best results have been obtained by Garmin device followed by Xiaomi Mi 8. These two devices have operational capability of more GNSS systems. The L5/E5 signals of the dual-frequency mobile phone receiver Xiaomi Mi 8 are more precise and less prone to distortions from multipath reflections [7]. The E5/L5 frequency makes it easier to distinguish real signals from the ones reflected by buildings, reducing the multipath effect, which is a major source of error [39]. In addition, the frequency diversity is more robust than interference and jamming. The better quality of the L5/E5 measurements may be a key to improving the navigation performance of the mobile phone receiver Xiaomi Mi 8.

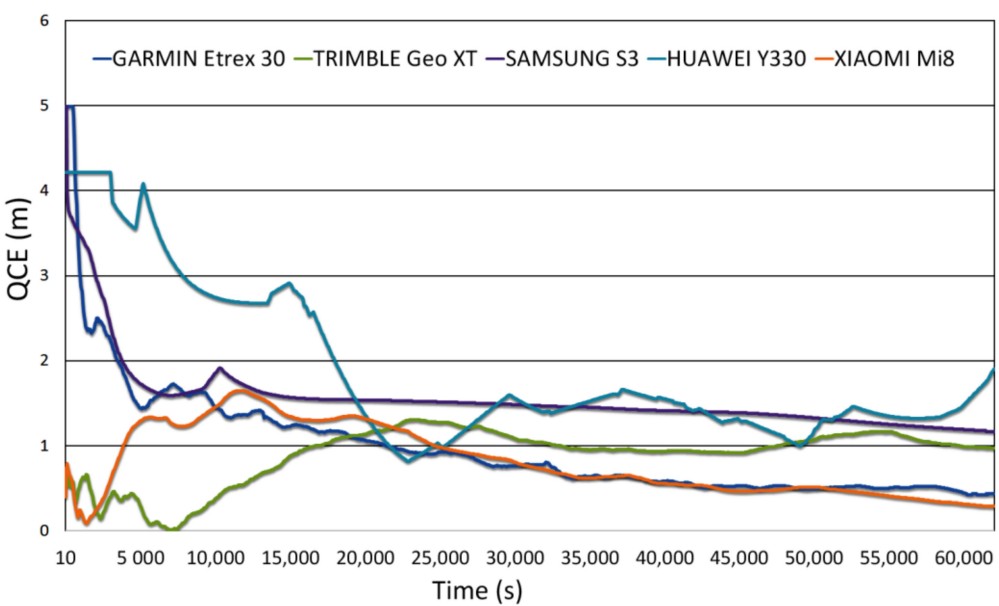

**Figure 2.** The relationship between the time and the quadratic composition of the East and North errors (QCE) of each device. From 10 to 60,000 s period of recorded data. East and North errors are the differences between the control point and coordinate means.

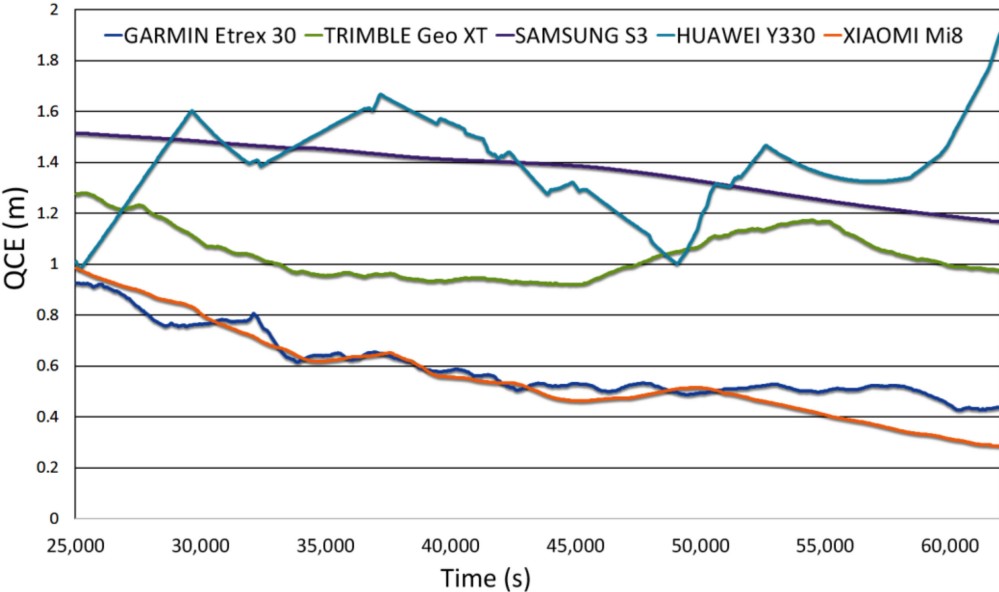

**Figure 3.** The relationship between the time and the quadratic composition of the East and North errors (QCE) of each device. From 25,000 to 60,000 s period of recorded data. East and North errors are the differences between the control point and coordinate means.

Garmin Etrex and Xiaomi Mi 8 had errors below the meter with a 25,000 s period of recorded data. More than 25,000 s did not improve the mean of the observations, and it became meaningless to continue measuring (see Tables A1 and A5 in Appendix A). With a 60,000 s period of observations, the Trimble Geo XT error was less than one meter. On the other hand, Huawei Y330-U01 and Samsung Galaxy S3 smartphones with 60,000 s did not achieve sub-meter accuracy.

With more than a 60,000 s period of time the error was maintained at the same level (see Tables A1–A5 and Figures A1–A5 in Appendix A). Thus, adding observations did not change the mean and no noticeable improvement in accuracy was seen in any device. There was no need to extend data collection over a 60,000 s period of recorded data.

About the different behavior displayed by the different devices, the smoothness of Xiaomi Mi 8, Samsung Galaxy S3 stands out. But we do not know the origin of that behavior.

### 3.1.1. Garmin Etrex 30 Measurements

The duration of GNSS Garmin recording was 2 months and, during that period the GNSS device did not move. The logging rate was set to 10 s.

### 3.1.2. Trimble Geo XT Measurements

The duration of GNSS Trimble Geo XT recording was 2 weeks, and, during this period the GNSS device was in stationary positions and taking a measurement every 10 s.

### 3.1.3. Smartphones Measurements: Samsung Galaxy S3, Huawei Y330-U01 and Xiaomi Mi 8

Regarding the smartphones (Samsung Galaxy S3, Huawei Y330-U01, and Xiaomi Mi 8) results, see Tables A4–A6. The duration of each recording was 32, 7, and 2.5 days and the frequency was 10, 10, and 1 s, respectively.

Therefore, we concluded that a large amount of data improved the accuracy of the mean measurements. Errors of several meters in open sky were reduced to sub-meter values.

### 3.2. Data Segmentation Test

In this case the whole data collection included 210,000 observations from the Xiaomi Mi 8 smartphone, with the logging rate set to 1 s. Different segments or groups were selected. Theoretically, as the amount of data increases, the quality of the coordinate mean improves and decreases the associated standard deviation (compare with Table 2).

**Table 2.** Segment type, segment size, number of segment, and associated East, North standard deviations, from Equation (1) with the $n$ values: 18, 11, 6, and 3.

| Segment | Segment Size (Observations) | Number of Segments $n$ | E Standard Deviation (m) | N Standard Deviation (m) |
|---------|------------------------------|-------------------------|---------------------------|---------------------------|
| 1 | 5000 | 18 | 0.85 | 1.05 |
| 2 | 20,000 | 11 | 0.32 | 0.26 |
| 3 | 40,000 | 6 | 0.19 | 0.13 |
| 4 | 70,000 | 3 | 0.23 | 0.14 |

From these segments, we obtained the mean for each element and the standard deviation (compare with Table 2).

An independent observation of the vertex allowed us to obtain the error and check the standard deviations. The results showed the real errors have a certain link with the calculated standard deviations; therefore, the segmentation process is only an approximate method.

### 3.3. Bootstrapping Test

Once we determined the segment mean and the associated variance (see Table 2), we used the bootstrapping method to check these results. Bootstrapping has been widely used and validated by statisticians and can be applied to arbitrary functions and data distributions [19].

R software was used in the case study research. The R package boot allows a user to easily generate bootstrap samples of virtually any statistic that they can calculate in R software.

According to Angrisano et al. [26], the $k$ value of thousands is sufficient; therefore, we adopted $k = 100,000$. The bootstrapping mean and standard deviation are contained in Table 3, which are similar to the standard deviations values of Table 2 from Equation (1), which proves that the bootstrapping standard deviations are a good estimate of the true standard deviations of the means obtained from the segmentation process.

**Table 3.** Segment type, number of segments k, East and North means, and the bootstrapping standard deviations associated.

| Segment | Number of Segments $n$ | k | E Mean (m) | N Mean (m) | E Standard Deviation (m) | N Standard Deviation (m) |
|---|---|---|---|---|---|---|
| 1 | 18 | 100,000 | 729,008.62 | 4,373,591.93 | 0.87 | 1.09 |
| 2 | 11 | 100,000 | 729,008.49 | 4,373,591.86 | 0.47 | 0.53 |
| 3 | 6 | 100,000 | 729,008.54 | 4,373,591.81 | 0.39 | 0.33 |
| 4 | 3 | 100,000 | 729,008.48 | 4,373,591.87 | 0.15 | 0.10 |

### 3.4. Network Adjustment

In this section the least squares method together with geometric constraints was tested in three different case studies. The first is a geodetic five points network (1) in very close positions using observations from the Xiaomi Mi 8 smartphone. This was placed on a sidewalk of the street. Errors of several dozens of meters in urban canyons are expected.

The second network (2) and third network (3) used different devices, forming part of a triangle. Distances between devices were approximately several meters. All these GNSS units were on the rooftop of a four-story building.

The final goal was to obtain the position and associated standard deviation of one vertex of the network. The mathematical model with its own conditions reinforced the confidence in the network results.

### 3.4.1. Five Points Network (1) Observed by the Xiaomi Mi 8 Smartphone

In this network the goal was to improve the position of the vertex number 1 (compare with Figure 4). A total of 2200 observations per vertex were collected successively from the Xiaomi Mi 8 smartphone for a 37 min period. The logging rate was set to 1 s. The mean of all these coordinates is the position of vertex 1 (compare with Table 4). The total number of observations from the five devices was 11,000 and, as we have seen in Table 2, the standard deviation expected from the statistical segmentation process was around 1 m.

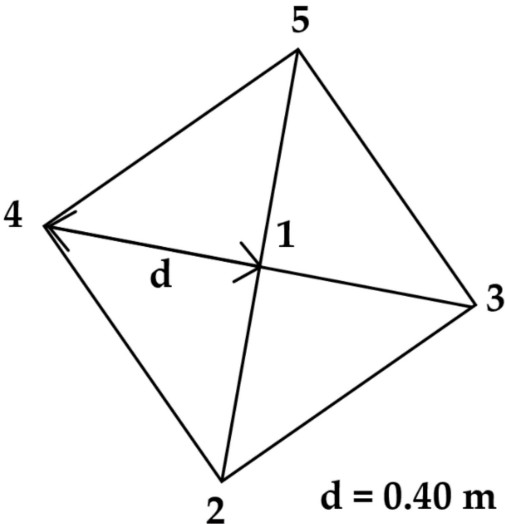

**Figure 4.** Vertex positions vertex 1, 2, 3, 4, and 5. The known distance from vertex 1 to 2, 3, 4, and 5 was 0.40 m.

**Table 4.** The East and North coordinates of vertex 1 obtained from all coordinate mean, from real-time GNSS techniques (E-RTK, N-RTK) and the differences between ($d_E$, $d_N$).

| Vertex | Smartphone | E Mean (m) | N Mean (m) | E-RTK (m) | N-RTK (m) | $d_E$ (m) | $d_N$ (m) |
|---|---|---|---|---|---|---|---|
| 1 | Xiaomi Mi 8 | 729,063.75 | 4,373,540.64 | 729,063.52 | 4,373,541.21 | −0.23 | 0.57 |

Here, we assume that the smartphones were mounted on a rigid platform, i.e., the relative distances between the smartphones remained unchanged. One GNSS device was chosen as the master device, in this case, the vertex number 1. The known distance from vertex 1 to 2, 3, 4, and 5 was 0.40 m.

We used 1, 4, and 5 vertices (see Figure 4) to form the network; therefore, the network looks like the networks in Sections 3.4.2 and 3.4.3.

The distances among receivers are fixed and known (d = 0.40 m). However, the mathematical model (27) requires coordinate increments. As we do not have prior information about the position of the XY/East-North axes, it is not possible to know the increments from 1 to, 4, and 5 vertices. Consequently, from the observed distances, virtual 4′ and 5′ coordinates were calculated derived from the master receptor/vertex 1 and receiver/vertices 4 and 5 for every observation.

$L_1$ is the vector of observed coordinates in the mathematical model (27). $L_1$ requires coordinates of vertices 1, 4′, and 5′. Figure 5 shows the point positions 1, 4′, and 5′.

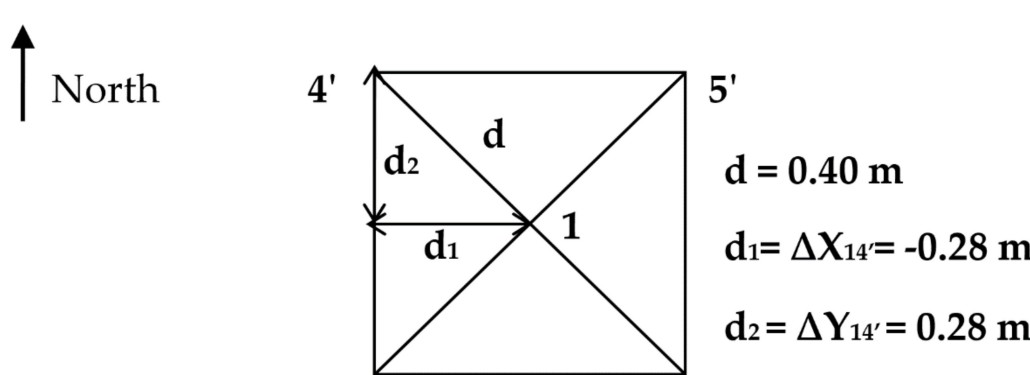

**Figure 5.** Vertex positions 1, 4′, and 5′. Distances d = $d_{14'}$ = $d_{15'}$ = 0.40 m. Coordinate increments: $d_1 = \Delta X_{14'} = -0.28$ m, $d_2 = \Delta Y_{14'} = 0.28$ m.

$L_2$ is the vector of coordinate of increments between vertices. $L_2$ has constant values and represents the geometric constraints. In this study case, Figure 4 shows the constant coordinates increments $x_{4'} - x_1 = -0.28$ m and $y_{4'} - y_1 = 0.28$ m.

Equation (27) allows us to obtain the adjusted x and y variables. The first step is to define all elements in Equation (27).

The coefficient matrix $A_1$:

$$A_1 = \begin{bmatrix} 1 & 0 & 0 \\ 0 & 1 & 0 \\ 0 & 0 & 1 \\ 1 & 0 & 0 \\ 0 & 1 & 0 \\ 0 & 0 & 1 \\ & . & \\ & . & \\ & . & \\ 1 & 0 & 0 \\ 0 & 1 & 0 \\ 0 & 0 & 1 \end{bmatrix}_{3nx3}$$

Vector *x* is:

$$x = \begin{bmatrix} x_1 \\ x_{4'} \\ x_{5'} \end{bmatrix}$$

Vector of independent terms $L_1$ as observed *x* coordinates $L_{1x}$:

Each observation will have its own $L_i$ vector, $i \in 1, 2, 3 \dots n$ observation. For example, the vector $L_{15}$ of observation 15 is:

$$L_{1x} = \begin{bmatrix} L_{11} \\ L_{12} \\ . \\ . \\ . \\ L_{1n} \end{bmatrix}_{3nx1}$$

$$L_{15} = \begin{bmatrix} 729063.75 \\ 729063.09 \\ 729065.22 \end{bmatrix}_{3x1}$$

The geometric constraints are established by the $A_2$ matrix as the second equation model. In this study case $A_2$ is:

$$x_1 - x_{5'} = -0.28$$

$$x_1 - x_{4'} = 0.28$$

and in matrix form:

$$A_2 \cdot x = L_2$$

with,

$$A_2 = \begin{bmatrix} 1 & 0 & -1 \\ 1 & -1 & 0 \end{bmatrix}$$

$$L_2 = \begin{bmatrix} -0.28 \\ 0.28 \end{bmatrix}$$

The Lagrange multiplier $\lambda_2$:

$$\lambda_2 = \begin{bmatrix} \lambda_{21} \\ \lambda_{22} \end{bmatrix}$$

We do not have any parameters that allow us to define the SPP observations weight matrix. Therefore, we computed the adjustments with equally weighted observations.

The mathematical model for $n$ Equation (27) can be expressed in the matrix form:

$$\begin{bmatrix} n & 0 & 0 & 1 & 1 \\ 0 & n & 0 & 0 & -1 \\ 0 & 0 & n & -1 & 0 \\ 1 & 0 & -1 & 0 & 0 \\ 1 & -1 & 0 & 0 & 0 \end{bmatrix} \cdot \begin{bmatrix} x_1 \\ x_{4'} \\ x_{5'} \\ \lambda_{21} \\ \lambda_{22} \end{bmatrix} = \begin{bmatrix} A_1^T \cdot L_{1x} \\ -0.28 \\ 0.28 \end{bmatrix}$$

The vector $x$ and Lagrange multipliers is then given by:

$$\begin{bmatrix} x_1 \\ x_{4'} \\ x_{5'} \\ \lambda_{21} \\ \lambda_{22} \end{bmatrix} = \begin{bmatrix} n & 0 & 0 & 1 & 1 \\ 0 & n & 0 & 0 & -1 \\ 0 & 0 & n & -1 & 0 \\ 1 & 0 & -1 & 0 & 0 \\ 1 & -1 & 0 & 0 & 0 \end{bmatrix}^{-1} \cdot \begin{bmatrix} A_1^T \cdot L_{1x} \\ -0.28 \\ 0.28 \end{bmatrix}$$

The process will be repeated with vector $y$ and Lagrange multipliers:

$$\begin{bmatrix} y_1 \\ y_{4'} \\ y_{5'} \\ \lambda_{23} \\ \lambda_{24} \end{bmatrix} = \begin{bmatrix} n & 0 & 0 & 1 & 1 \\ 0 & n & 0 & 0 & -1 \\ 0 & 0 & n & -1 & 0 \\ 1 & 0 & -1 & 0 & 0 \\ 1 & -1 & 0 & 0 & 0 \end{bmatrix}^{-1} \cdot \begin{bmatrix} A_1^T \cdot L_{1y} \\ -0.28 \\ -0.28 \end{bmatrix}$$

In Table 5, you can find the coordinate means from all the data collected.

**Table 5.** Vertices 1, 4′, and 5′: East, North means of data collected, East, North coordinates from real-time GNSS techniques (E-RTK, N-RTK), and the differences between ($d_E$, $d_N$).

| Vertex | Smartphone | E Mean (m) | N Mean (m) | E-RTK (m) | N-RTK (m) | $d_E$ (m) | $d_N$ (m) |
|---|---|---|---|---|---|---|---|
| 1 | Xiaomi Mi 8 | 729,063.8 | 4,373,541 | 729,063.5 | 4,373,541 | −0.23 | 0.57 |
| 4′ | Xiaomi Mi 8 | 729,063.1 | 4,373,541 | 729,063.2 | 4,373,541 | 0.15 | 0.18 |
| 5′ | Xiaomi Mi 8 | 729,065.2 | 4,373,542 | 729,063.8 | 4,373,541 | −1.42 | −0.61 |

Table 6 shows the differences between the adjusted coordinates obtained from Equation (27) and the real position of each vertex obtained using real-time GNSS techniques. The efficiency of the model from Equation (27) was evaluated by comparing the results of Tables 5 and 6. We would like to note that adding constraints provided a better solution, and point 1 improved the position values from the referenced points.

**Table 6.** Vertices 1, 4′, and 5′: East, North coordinates obtained from Equation (27), East, North coordinates from real-time GNSS techniques (E-RTK, N-RTK), and the differences between ($d_E$, $d_N$).

| Vertex. | Smartphone | E Mean (m) | N Mean (m) | E-RTK (m) | N-RTK (m) | $d_E$ (m) | $d_N$ (m) |
|---|---|---|---|---|---|---|---|
| 1 | Xiaomi Mi 8 | 729,064 | 4,373,541 | 729,063.5 | 4,373,541 | −0.5 | 0.04 |
| 4′ | Xiaomi Mi 8 | 729,063.7 | 4,373,541 | 729,063.2 | 4,373,541 | −0.5 | 0.04 |
| 5′ | Xiaomi Mi 8 | 729,064.3 | 4,373,541 | 729,063.8 | 4,373,541 | −0.5 | 0.04 |

The position of the antenna phase center is unknown, but is not a problem, because the quadratic composition of the network errors: $Qc = \sqrt{d_E^2 + d_N^2} = \sqrt{-0.50^2 + 0.04^2} = 0.502$ m (compare with Table 6) is higher than the tolerance of Xiaomi smartphone $T_{Xiaomi} = 0.172$ m (compare with Table 1).

As for the accuracy of the observations we are using the expression of the variance–covariance Equation (35), where:

$$\sigma_0^2 = \frac{v_1^T \cdot P_1 \cdot v_1}{n_1 + n_2 - u} = 0.67 \ m^2$$

$$Q_x = Q_{x^*} + \Delta Q_x = \begin{pmatrix} 1 & 0 & 0 \\ 0 & 1 & 0 \\ 0 & 0 & 1 \end{pmatrix} + \begin{pmatrix} -0.667 & 0.333 & 0.333 \\ 0.333 & -0.667 & 0.333 \\ 0.333 & 0.333 & -0.667 \end{pmatrix}$$
$$= \begin{pmatrix} 0.333 & 0.333 & 0.333 \\ 0.333 & 0.333 & 0.333 \\ 0.333 & 0.333 & 0.333 \end{pmatrix}$$

$$\Sigma_x = \sigma_0^2 \cdot Q_x = \begin{pmatrix} 0.2231 & 0.2231 & 0.2233 \\ 0.2231 & 0.2231 & 0.2233 \\ 0.2231 & 0.2231 & 0.2233 \end{pmatrix}$$

According to the matrix variance-covariance $\Sigma_x$, it will be a standard deviation value of 0.47 m in X/East coordinates.

The same process gives us the Y/North coordinates standard deviation.

$$\sigma_0^2 = \frac{v_1^T \cdot P_1 \cdot v_1}{n_1 + n_2 - u} = 0.36 \ m^2$$

$$\Sigma_y = \sigma_0^2 \cdot Q_x = \begin{pmatrix} 0.1213 & 0.1213 & 0.1213 \\ 0.1213 & 0.1213 & 0.1213 \\ 0.1213 & 0.1213 & 0.1213 \end{pmatrix}$$

with a standard deviation of 0.35 m in y coordinates.

### 3.4.2. Three Points Network (2) Observed by Three Different GNSS Devices

The input data consists of an observation set of three vertices number 1, 2 and 3 (see (b) right Figure 1). A total of 51,330 coordinates per vertex were collected from several GNSS devices simultaneously: on vertex 1 Huawei Y330-U01, on vertex 2 Trimble Geo XT, and on vertex 3 Garmin Etrex 30. The logging rate was set to 10 s. The network size was far greater than the network (1).

Following essentially the same methodology of Section 3.4.1, Tables 7 and 8 describes the results obtained. Table 7 shows the differences between the means of the 51,330 observations and the real position of each vertex, whose values come from real-time GNSS techniques.

**Table 7.** Vertices 1, 2, and 3: East, North means of data collected, East, North coordinates from real-time GNSS techniques (E-RTK, N-RTK) and the differences between ($d_E$, $d_N$).

| Vertex | Device | E Mean (m) | N Mean (m) | E-RTK (m) | N-RTK (m) | $d_E$ (m) | $d_N$ (m) |
|--------|--------|-----------|-----------|-----------|-----------|-----------|-----------|
| 1 | Huawei | 729,009.2 | 4,373,593 | 729,008.8 | 4,373,592 | −0.37 | −1.02 |
| 2 | Trimble GeoXT | 729,014.6 | 4,373,595 | 729,014.2 | 4,373,594 | −0.42 | −1.05 |
| 3 | Garmin Etrex | 729,010.1 | 4,373,582 | 729,010 | 4,373,582 | −0.15 | 0.66 |

**Table 8.** Vertices 1, 2, and 3: East, North coordinates obtained from Equation (27), East, North coordinates from real-time GNSS techniques (E-RTK, N-RTK) and the differences between ($d_E$, $d_N$).

| Vertex | Device | E Mean (m) | N Mean (m) | E-RTK (m) | N-RTK (m) | $d_E$ (m) | $d_N$ (m) |
|--------|--------|-----------|-----------|-----------|-----------|-----------|-----------|
| 1 | Huawei | 729,009.1 | 4,373,592 | 729,008.8 | 4,373,592 | −0.31 | −0.47 |
| 2 | Trimble Geo XT | 729,014.5 | 4,373,595 | 729,014.2 | 4,373,594 | −0.31 | −0.47 |
| 3 | Garmin Etrex | 729,010.3 | 4,373,583 | 729,010 | 4,373,582 | −0.31 | −0.47 |

Table 8 shows the differences between the adjusted coordinates obtained from Equation (27) and the real position of each vertex obtained using real-time GNSS techniques.

The efficiency of the model from the Equation (27) was evaluated by comparing the results of Tables 7 and 8. The constraints may improve the accuracy, from $d_E = -0.37$ m and $d_N = -1.02$ m to $d_E = -0.31$ m and $d_N = -0.47$ m (see vertex 1 errors in Tables 7 and 8). Xiaomi tolerance is $T_{Xiaomi} = 0.172$ m; therefore, not knowing the position of the antenna is not a problem.

There is one further advantage in applying conditions in the adjusted least squares system. A surprising finding was that it was seldom necessary to use the whole dataset, and 50% of the data was sufficient to achieve similar results compared to the execution over the entire dataset. This required fewer observations because the system converged earlier, as can be seen comparing Tables 9 and 10. The error values were similar with 25,000 observations than with 51,330, making more efficient use of the available resources by ceasing to process data when the confidence region error was sufficiently small.

Table 9 shows the $d_E$, $d_N$ error of different periods of observations, ranking in increasing order. Table 10 shows the $d_E$, $d_N$ error of observations adjusted by Equation (27).

According to the matrix variance-covariance $\Sigma_x$ (see Equation (35)), there will be a standard deviation value of 0.08 m in E coordinates and 0.57 m in N coordinates.

**Table 9.** The $d_E$, $d_N$ error from the East, North coordinates obtained from the means of different number of observation.

| Observations | Vertex 1 $d_E$ (m) | Vertex 1 $d_N$ (m) | Vertex 2 $d_E$ (m) | Vertex 2 $d_N$ (m) | Vertex 3 $d_E$ (m) | Vertex 3 $d_N$ (m) |
|---|---|---|---|---|---|---|
| 5000 | −0.3 | −1.12 | −1.12 | −0.28 | 0.09 | 0.49 |
| 10,000 | −2.17 | −1.05 | −1.05 | −0.3 | 0.03 | 0.39 |
| 15,000 | −1.43 | −0.79 | −0.79 | −0.28 | 0.03 | 0.76 |
| 20,000 | −1.07 | −0.81 | −0.81 | −0.27 | −0.03 | 0.8 |
| 25,000 | −0.42 | −1.06 | −1.06 | −0.4 | −0.14 | 0.62 |
| 30,000 | −0.74 | −1.06 | −1.06 | −0.34 | −0.11 | 0.55 |
| 35,000 | −0.6 | −1.14 | −1.14 | −0.33 | −0.14 | 0.44 |
| 40,000 | −0.59 | −1.18 | −1.18 | −0.32 | −0.11 | 0.48 |
| 45,000 | −0.43 | −1.05 | −1.05 | −0.36 | −0.11 | 0.51 |
| 51,330 | −0.42 | −1.06 | −1.06 | −0.4 | −0.14 | 0.62 |

**Table 10.** The $d_E$, $d_N$ error from the East, North coordinates obtained from Equation (27) of different number of observation.

| Observations | Vertex 1 $d_E$ (m) | Vertex 1 $d_N$ (m) | Vertex 2 $d_E$ (m) | Vertex 2 $d_N$ (m) | Vertex 3 $d_E$ (m) | Vertex 3 $d_N$ (m) |
|---|---|---|---|---|---|---|
| 5000 | −0.16 | −0.63 | −0.16 | −0.63 | −0.16 | −0.6 |
| 10,000 | −0.81 | −0.57 | −0.81 | −0.57 | −0.81 | −0.54 |
| 15,000 | −0.56 | −0.37 | −0.56 | −0.37 | −0.56 | −0.34 |
| 20,000 | −0.45 | −0.35 | −0.45 | −0.35 | −0.45 | −0.32 |
| 25,000 | −0.31 | −0.51 | −0.32 | −0.51 | −0.32 | −0.48 |
| 30,000 | −0.39 | −0.51 | −0.39 | −0.51 | −0.39 | −0.48 |
| 35,000 | −0.35 | −0.59 | −0.35 | −0.59 | −0.35 | −0.56 |
| 40,000 | −0.34 | −0.6 | −0.34 | −0.6 | −0.34 | −0.57 |
| 45,000 | −0.3 | −0.54 | −0.3 | −0.54 | −0.3 | −0.51 |
| 51,330 | −0.31 | −0.51 | −0.32 | −0.51 | −0.32 | −0.48 |

### 3.4.3. Three Points Network (3) Observed by Xiaomi Mi 8

The input data consists of an observation set of vertices 1, 2 and 3 (see (b) right Figure 1). A total of 1493 coordinates per vertex were collected successively from the Xiaomi Mi 8 smartphone. The logging rate was set to 1 s. The network size was far greater than the network of Section 3.4.1.

Table 11 shows the differences between the means of the 1493 observations and the real position of each vertex, where the values come from real-time GNSS techniques. Table 12 shows the differences between the adjusted coordinates obtained from Equation (27) and the real position of each vertex obtained using real-time GNSS techniques.

**Table 11.** Vertices 1, 2, and 3: East, North means of data collected, East, North coordinates from real-time GNSS techniques (E-RTK, N-RTK) and the differences between ($d_E$, $d_N$).

| Vertex | Smartphone | E Mean (m) | N Mean (m) | E-RTK (m) | N-RTK (m) | $d_E$ (m) | $d_N$ (m) |
|---|---|---|---|---|---|---|---|
| 1 | Xiaomi Mi 8 | 729,008.1 | 4,373,592 | 729,008.8 | 4,373,592 | 0.66 | −0.46 |
| 2 | Xiaomi Mi 8 | 729,014.2 | 4,373,594 | 729,014.2 | 4,373,594 | −0.09 | 0.53 |
| 3 | Xiaomi Mi 8 | 729,008.5 | 4,373,583 | 729,010 | 4,373,582 | 1.51 | −0.57 |

**Table 12.** Vertices 1, 2, and 3: East, North coordinates obtained from expression (27), East, North coordinates from real-time GNSS techniques (E-RTK, N-RTK) and the differences between ($d_E$, $d_N$).

| Vertex | Smartphone | E Mean (m) | N Mean (m) | E-RTK (m) | N-RTK (m) | $d_E$ (m) | $d_N$ (m) |
|---|---|---|---|---|---|---|---|
| 1 | Xiaomi Mi 8 | 729,008.1 | 4,373,592 | 729,008.8 | 4,373,592 | 0.69 | −0.17 |
| 2 | Xiaomi Mi 8 | 729,013.5 | 4,373,594 | 729,014.2 | 4,373,594 | 0.69 | −0.17 |
| 3 | Xiaomi Mi 8 | 729,009.3 | 4,373,583 | 729,010 | 4,373,582 | 0.69 | −0.17 |

According to the matrix variance-covariance $\Sigma_x$ (see Equation (35)), there will be a standard deviation value of 0.32 metres in E coordinates and 0.19 m in N coordinates.

As in Section 3.4.2, fewer observations were needed to obtain the same result, because the system converges earlier, as can be seen in comparing Tables A6 and A7 in the Appendix A. This is a way to reduce the convergence time so that 27 min are needed to collect 1500 observations. This is a short period of time and coordinate errors were around 0.50 m.

## 4. Discussion

Hereafter, the main observations based on the experiments conducted are explained. The first simple question was if a superior quantity of GNSS data would improve the accuracy of the measurements' mean. Therefore, we analyzed and assessed the positioning accuracy of a large number of static observations collected by different GNSS devices. The GNSS operational mode used was the single point positioning (SPP), which can reach a positioning accuracy around 5–10 m in the open sky. SPP coordinates were collected to obtain the averages. These averages were compared with the true position achieved by a geodetic receiver. However, the true coordinates of the low-cost devices are unknown, as the exact position of the antennas are unknown. The assumption made was to consider the tolerance for the precision of positioning as the size of each GNSS device. All tolerance values were below 0.50 m.

With more than a 60,000 s period of recorded data, the error was maintained at the same level (see Tables A1–A5 and Figures A1–A5 in Appendix A). Thus, adding observations did not change the mean, and no noticeable improvement in accuracy was seen.

The best results were obtained using the Garmin Etrex 30 and Xiaomi Mi 8 smartphone. From 40,000 s period of time in the open sky, the quadratic components of the E, N errors were $Qc_{GARMIN} = 0.41$ m and $Qc_{Xiaomi} = 0.56$ m. With 25,000 s a sub-meter horizontal accuracy was achieved (compare with Figure 2).

On the other hand, the Huawei Y330-U01 and Samsung Galaxy S3 smartphones with 60,000 s period of recorded data did not achieve sub-meter accuracy.

The results were not surprising as the Xiaomi Mi 8 smartphone represents one of the most recent advances and developments of GNSS. An important improvement is the introduction of a dual frequency chipset, which on top of L1/E1, will provide the GPS L5 and Galileo E5 signals that are much more resistant to multipath error and could eliminate the ionospheric error. Tracking multiple GNSS constellations maximizes the availability of a position fix even in harsh environments such as urban canyons [6] and this is another advantage. To obtain a reliable positioning in the case of limited sky visibility, common in urban areas, this can only be achieved using a multi-GNSS solution, which Xiaomi Mi 8 provided as well. While the minimum satellite requirement is four, to obtain a reliable positioning, especially in the presence of noise and obstruction, this requires 8–10 satellites [40].

To estimate the quality of the data, segmentation of the observations is proposed. If the size of the segment increases, the standard deviation decreases, as was expected. After the segmentation process, the bootstrapping method was used to check and confirm the standard deviations obtained in segmentation process. However, as was said in Section 3.2, real errors only have a certain link with the calculated standard deviations, consequently the segmentation process is only an approximate method.

The large amount of static observations collected by low-cost GNSS devices may achieve a sub-meter horizontal accuracy. However, the drawback is the long period of time required. A least squares solution network was applied over all the recorded data, assisted by external geometric conditions. More sensors may provide more data in less time, and so the period of time was reduced. We proposed a five-vertices GNSS network (see (a) left Figure 1). These five points of network (1) gave us a total of 11,000 observations. The value of vertex 1 was the average of these 11,000 observations, and the quadratic component of the East, North errors was $QcM_{Xiaomi} = 0.61$ m (compare with Table 13).

**Table 13.** The quadratic component QcM of the E, N errors from the mean of the observations. The quadratic component QcAd of the E, N errors from the adjusted coordinates. The quadratic component $Qc\sigma_{EN}$ of $\sigma_E$ and $\sigma_N$ from expresScheme 35.

| Vertex 1 | Observations per Vertex | Time (Minutes) | QcM (m) | QcAd (m) | $Qc\sigma_{EN}$ (m) |
|---|---|---|---|---|---|
| **Network (1)** | 2200 | 37 | 0.61 | 0.50 | 0.58 |
| **Network (2)** | 25,000 | 4166 | 1.08 | 0.56 | 0.57 |
| **Network (3)** | 1493 | 25 | 0.80 | 0.51 | 0.37 |

After, the least squared method together with geometric constraints provided the adjusted coordinates and the quadratic component QcAd of the East, North errors: $QcAd_{Xiaomi} = 0.50$ m (compare with Table 13).

A similar process in was used with (2) and (3) networks. Network (2) required much more time. The logging rate was set to 10 s and the observations collected totaled 25,000; therefore, the convergence time was three days. Measurements were taken by several GNSS devices, the Huawei Y330-U01, Trimble Geo XT, and Garmin Etrex 30 device.

Table 13 shows the results of the observations' mean and the adjusted networks, of the master receiver. As can be seen, the least-squares adjustments had fewer errors.

Least-squares adjustment is most often associated with high precision surveying. However, least-squares adjustment is a device for carrying out objective quality control of measurements by processing sets of redundant observations using the expression of the variance–covariance matrix (35). The variances of vertex 1 (compare with Table 13) were consistent with the real errors. This was not the case for the variance obtained from data segmentation.

Network (1) was on the walk side of the street in an urban area, and networks (2) and (3) were on the rooftop of a four-story building. The distances between devices were 0.40 m in network (1) but around several meters in networks (2) and (3).

A surprising finding was that it was seldom necessary to use the whole dataset if we used Equation (28). Networks (2) and (3) demonstrated that it was sufficient with 50–70% of the data to achieve similar results compared to the execution over the entire dataset. For instance, the error values of network (2) were similar with 25,000 s period of recorded data and with 51,330. Thus, we can reduce the period of time.

## 5. Conclusions

In accordance with Mayer-Schönberger and Cukier [21], a world of big data will require us to change our thinking about the value of accuracy. Exactness requires carefully curated data, small quantities, and, of course, certain situations still require it. However, to use the conventional mindset of measurement in the digital, connected world of the present century is to lose out on a crucial opportunity. Looking at vastly more data also permits us to loosen up the instrumental accuracy

The typical performance of today's mass-market GNSS devices is in the range of meters to even tens of meters in difficult conditions, such as urban canyons [40]. The most common GNSS operational mode is the single point positioning (SPP), related to pseudorange measurements. This mode limits their level of accuracy to several meters in open sky and to several dozens of meters in urban canyons. The present paper proposes to improve the SPP measurements for mass-market devices looking for sub-meter accuracy in static positions, by the process of recording GNSS observations for a long period of time. To reduce the satellite tracking time and to control the quality of measurements, we applied a least squares solution network, assisted by external geometric conditions. The completed process was tested with examples. Prominent among them was a network using a dual-frequency Xiaomi Mi 8 smartphone in an urban area, with 0.50 m of horizontal accuracy in 37 min. This is an alternative proposal to obtain a sub-meter accuracy level.

As this paper describes, one device equipped with several GNSS chipsets, which allows all components to be housed on a chassis, may achieve sub-meter accuracy. The size of the GNSS chipsets, with regard to other geodetic GNSS receivers, allows it to be located

in different spaces, objects, or places. The final goal is to obtain the absolute positioning and associated standard deviations of one vertex from receivers positioned on a network.

An advantage of working with these inexpensive receivers is in not requiring external corrections or high-precision products, especially when internet is not available. The SPP mode is independent from any local deformation or movement from the reference stations.

Smartphone measurements are highly noisy. The pseudorange noise of smart devices is about 10 times larger than that of geodetic receivers. Carrier phase observations are a major obstacle for smart devices to achieve precision positioning [3]. However, present and future research in GNSS devices should contain carrier phase observations to find the absolute position from mass-market GNSS receivers.

Many mass-market applications can benefit from increased accuracy. Dual frequency smartphones can meet the requirements of most application scenarios and can be used in semiprofessional yields [8], like augmented reality tools in mobile applications, map updating or cadastral survey. An example can be found in the construction industry. The industry is moving towards using augmented reality to visualise georeferenced models of construction sites, underground structures, cables, and pipes using mobile devices [41]. Likewise, the Internet of Things, which allows physical devices, vehicles, buildings, and other objects to be inter-connected and remotely controlled by GNSS localization, creating its own infrastructure. The pervasive presence of a variety of objects which can interact with each other and cooperate with their neighbors for reaching common goals [1]. Smart lighting or garbage management are just a couple of examples of applications that might benefit from improved accuracy in smart city asset management, or in areas where the access may be difficult, such as archaeological ones [39]. Mobile phones and mobile technology could be deployed as a grid of sensors, which can be used to control slow movement speeds, like a glacier, a mass of snow and ice that moves slowly away from carving gradually a broad and steep-sided valley on its way. The 3GPP (3rd Generation Partnership Project) technologies, such as the radio standard narrowband internet of things (NB-IoT), are suitable for communications but cannot provide reasonable accuracies (<50 m). Hybridisation of optimised GNSS for low-power together with 3GPP technologies for comms may provide the right balance. At present, mobile technology can contain elements for storing data, RAM, and advanced microprocessors, and the recorded information can be transferred using mobile Internet, Wi-Fi, or Bluetooth connectivity.

SPP mode is often combined with other information sources, such as maps, inertial sensors, or vision systems [39,42,43]. The literature reported the usefulness of fusing GPS, accelerometer, and smartphone information to discriminate undesired noise, which can be generated by atmospheric disturbances [44]. Accelerometers and GNSS complement each other [45–47]. External information from cheap sensors promises to increase the GNSS accuracy, and so researchers have proposed a GNSS/IMU/odometry integration algorithm [48]. The study of positioning techniques based on sensor fusion could provide actionable information in the present and near future. Hybridization with other technologies, as G5 may complement GNSS towards environment-independent use.

Finally, we would like to emphasize that an important source of big data will come from mobile phones and other devices. The information will be less accurate; however, the great volume makes it worthwhile to forgo strict exactitude. The use of low-cost double-frequency receivers will become very competitive in comparison to their more expensive cousins, and thus, in surveying operations with significantly lower accuracies, they could substitute for complex and expensive instruments, such as traditional GNSS receivers [49].

**Author Contributions:** Conceptualization, M.J.J.-M.; methodology, M.J.J.-M.; software, M.J.J.-M.; validation, M.J.J.-M., M.F.-A. and N.Q.-O.; formal analysis, M.J.J.-M., M.F.-A. and N.Q.-O.; investigation, M.J.J.-M.; resources, M.J.J.-M., M.F.-A. and N.Q.-O.; data curation, M.J.J.-M., M.F.-A. and N.Q.-O.; writing—original draft preparation, M.J.J.-M.; writing—review and editing, M.J.J.-M., M.F.-A. and N.Q.-O.; visualization, M.J.J.-M.; supervision, M.J.J.-M., M.F.-A. and N.Q.-O.; project administration, M.J.J.-M. and M.F.-A.; funding acquisition, M.F.-A. All authors have read and agreed to the published version of the manuscript.

**Funding:** This research was funded by the Comunidad de Madrid (CAM) and the European Social Fund (ESF) grant number [H2019/HUM-5742 AVIPES-CM.

**Institutional Review Board Statement:** Not applicable.

**Informed Consent Statement:** Not applicable.

**Data Availability Statement:** Not applicable.

**Acknowledgments:** The authors are grateful to the Editor and anonymous reviewers for their valuable suggestions and constructive comments. They have provided key insights and helped the authors shape the early drafts of the paper into its final form. The present work was carried out within the project H2019/HUM-5742 AVIPES-CM financed by the Comunidad de Madrid (CAM) and the European Social Fund (ESF).

**Conflicts of Interest:** The authors declare no conflict of interest. The founding sponsors had no role in the design of the study; in the collection, analyses, or interpretation of data; in the writing of the manuscript, and in the decision to publish the results.

## Appendix A

From five different devices, Tables A1–A5 show the period of recorded data and the mean coordinates of East and North. Differences between the reference data and coordinate means (E-DRTK and N-DRTK). Tables A1–A5 have been plotted in Figures A1–A5.

Table A6 shows the $d_E$, $d_N$ error of different periods of observations in network (3), ranking in increasing order. Table A7, shows the $d_E$, $d_N$ error of observations adjusted by expression (27) in network (3).

**Table A1.** Garmin Etrex 30 device period of recorded data, and the coordinate means for E and N. Differences between the reference data and coordinates mean (E-DRTK and N-DRTK). Real-time techniques were used to obtain the reference data: 729,009.945 m, 4,373,582.379 m.

| Period of Recorded Data (s) | E Mean (m) | N Mean (m) | E-DRTK (m) | N-DRTK (m) |
|---|---|---|---|---|
| 350,980 | 729,009.8 | 4,373,582 | 0.16 | 0.454 |
| 430,660 | 729,009.9 | 4,373,582 | 0.088 | 0.392 |
| 505,140 | 729,009.8 | 4,373,582 | 0.175 | 0.223 |
| 594,580 | 729,009.8 | 4,373,582 | 0.139 | 0.185 |
| 676,220 | 729,009.8 | 4,373,582 | 0.112 | 0.116 |
| 757,240 | 729,009.8 | 4,373,582 | 0.134 | −0.032 |
| 866,780 | 729,009.8 | 4,373,582 | 0.11 | −0.116 |
| 942,970 | 729,009.8 | 4,373,583 | 0.102 | −0.142 |
| 1,340,830 | 729,009.8 | 4,373,582 | 0.109 | 0.381 |
| 1,422,310 | 729,009.8 | 4,373,583 | 0.109 | −0.381 |
| 1,611,310 | 729,009.8 | 4,373,583 | 0.18 | −0.417 |
| 1,699,930 | 729,009.8 | 4,373,583 | 0.169 | −0.425 |
| 1,872,400 | 729,009.8 | 4,373,583 | 0.145 | −0.482 |
| 1,999,440 | 729,009.8 | 4,373,583 | 0.136 | −0.484 |
| 2,154,670 | 729,009.8 | 4,373,583 | 0.135 | −0.505 |
| 2,706,200 | 729,009.8 | 4,373,583 | 0.111 | −0.433 |
| 2,808,710 | 729,009.8 | 4,373,583 | 0.111 | −0.433 |
| 3,777,260 | 729,009.8 | 4,373,583 | 0.112 | −0.411 |
| 3,831,260 | 729,009.8 | 4,373,583 | 0.111 | −0.41 |
| 3,909,810 | 729,009.8 | 4,373,583 | 0.113 | −0.434 |
| 3,963,810 | 729,009.8 | 4,373,583 | 0.097 | −0.386 |
| 4,017,810 | 729,009.8 | 4,373,583 | 0.096 | −0.372 |
| 4,092,710 | 729,009.8 | 4,373,583 | 0.097 | −0.375 |
| 4,173,710 | 729,009.8 | 4,373,583 | 0.096 | −0.374 |

**Table A1.** *Cont.*

| Period of Recorded Data (s) | E Mean (m) | N Mean (m) | E-DRTK (m) | N-DRTK (m) |
|---|---|---|---|---|
| 4,254,710 | 729,009.9 | 4,373,583 | 0.095 | −0.377 |
| 4,391,420 | 729,009.9 | 4,373,583 | 0.094 | −0.388 |
| 4,463,090 | 729,009.9 | 4,373,583 | 0.091 | −0.39 |
| 4,684,810 | 729,009.8 | 4,373,583 | 0.097 | −0.39 |
| 4,738,810 | 729,009.8 | 4,373,583 | 0.097 | −0.39 |
| 4,829,140 | 729,009.8 | 4,373,583 | 0.1 | −0.382 |
| 4,856,140 | 729,009.8 | 4,373,583 | 0.098 | −0.38 |
| 4,883,140 | 729,009.8 | 4,373,583 | 0.097 | −0.378 |
| 4,910,140 | 729,009.8 | 4,373,583 | 0.096 | −0.378 |
| 4,937,140 | 729,009.9 | 4,373,583 | 0.093 | −0.382 |
| 4,964,140 | 729,009.9 | 4,373,583 | 0.092 | −0.379 |
| 5,057,220 | 729,009.9 | 4,373,583 | 0.093 | −0.379 |
| 5,084,220 | 729,009.9 | 4,373,583 | 0.092 | −0.379 |
| 5,180,640 | 729,009.9 | 4,373,583 | 0.093 | −0.379 |

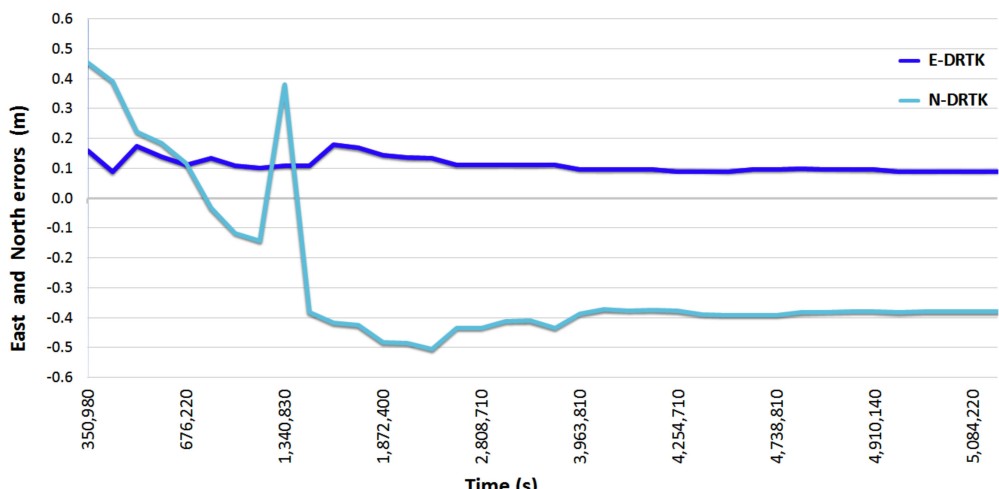

**Figure A1.** The relationship between the time and the East and North errors of Garmin Etrex 30 device. East and North errors are the differences between the control point and coordinate means.

**Table A2.** Trimble Geo XT device period of recorded data, and coordinate means for E and N. Differences between the reference data and coordinate means (E-DRTK and N-DRTK). Real-time techniques were used to obtain the reference data: 729,014.1534 m, 4,373,594.110 m.

| Period of Recorded Data (s) | E Mean (m) | N Mean (m) | E-DRTK (m) | N-DRTK (m) |
|---|---|---|---|---|
| 100,000 | 729,014.4 | 4,373,595 | −0.245 | −0.934 |
| 200,000 | 729,014.5 | 4,373,595 | −0.315 | −0.964 |
| 300,000 | 729,014.5 | 4,373,595 | −0.388 | −0.947 |
| 400,000 | 729,014.5 | 4,373,595 | −0.374 | −0.965 |
| 500,000 | 729,014.5 | 4,373,595 | −0.373 | −0.992 |
| 600,000 | 729,014.5 | 4,373,595 | −0.374 | −0.935 |
| 700,000 | 729,014.5 | 4,373,595 | −0.395 | −0.959 |
| 800,000 | 729,014.5 | 4,373,595 | −0.393 | −0.936 |
| 900,000 | 729,014.5 | 4,373,595 | −0.376 | −0.932 |
| 1,000,000 | 729,014.5 | 4,373,595 | −0.37 | −0.904 |
| 1,100,000 | 729,014.5 | 4,373,595 | −0.362 | −0.881 |
| 1,245,360 | 729,014.5 | 4,373,595 | −0.358 | −0.872 |

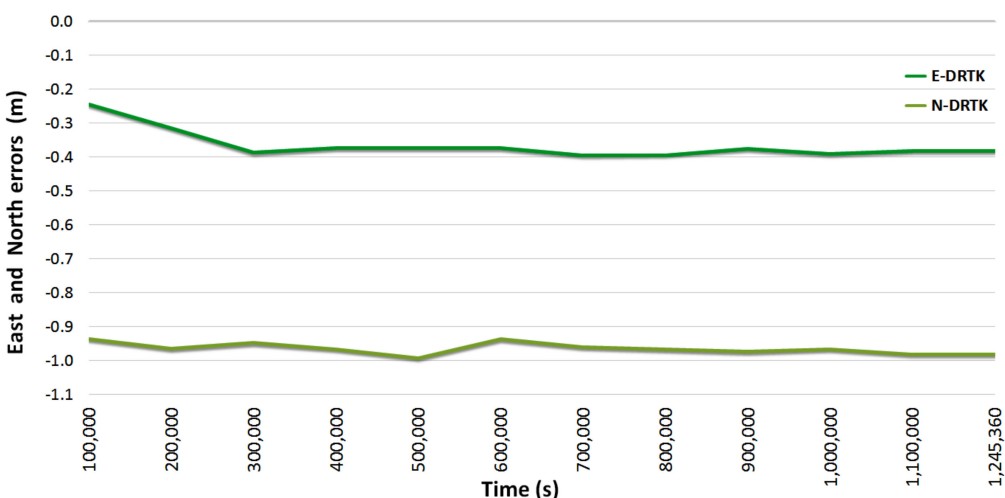

**Figure A2.** The relationship between the time and the East and North errors of Trimble Geo XT device. East and North errors are the differences between the control point and coordinate means.

**Table A3.** Samsung Galaxy S3 device period of recorded data, and coordinate means for E and N. Differences between the reference data and coordinate means (E-DRTK and N-DRTK). Real-time techniques were used to obtain the reference data: 729,014.153 m, 4,373,594.11 m.

| Period of Recorded Data (s) | E Mean (m) | N Mean (m) | E-DRTK (m) | N-DRTK (m) |
|---|---|---|---|---|
| 100,000 | 729,013.3 | 4,373,595 | 0.873 | −0.67 |
| 200,000 | 729,013.4 | 4,373,595 | 0.708 | −0.949 |
| 300,000 | 729,013.5 | 4,373,595 | 0.639 | −0.919 |
| 400,000 | 729,013.6 | 4,373,595 | 0.597 | −0.988 |
| 500,000 | 729,013.6 | 4,373,595 | 0.602 | −0.996 |
| 600,000 | 729,013.5 | 4,373,595 | 0.611 | −1.047 |
| 700,000 | 729,013.5 | 4,373,595 | 0.651 | −1.097 |
| 800,000 | 729,013.5 | 4,373,595 | 0.679 | −1.107 |
| 900,000 | 729,013.5 | 4,373,595 | 0.66 | −1.23 |
| 1,000,000 | 729,013.5 | 4,373,595 | 0.637 | −1.248 |
| 1,100,000 | 729,013.5 | 4,373,595 | 0.631 | −1.242 |
| 1,200,000 | 729,013.5 | 4,373,595 | 0.633 | −1.289 |
| 1,300,000 | 729,013.5 | 4,373,595 | 0.634 | −1.305 |
| 1,400,000 | 729,013.5 | 4,373,595 | 0.628 | −1.287 |
| 1,500,000 | 729,013.5 | 4,373,595 | 0.64 | −1.295 |
| 1,600,000 | 729,013.5 | 4,373,595 | 0.648 | −1.294 |
| 1,700,000 | 729,013.5 | 4,373,595 | 0.67 | −1.307 |
| 1,800,000 | 729,013.5 | 4,373,595 | 0.685 | −1.275 |
| 1,900,000 | 729,013.4 | 4,373,595 | 0.726 | −1.255 |
| 2,000,000 | 729,013.4 | 4,373,595 | 0.754 | −1.245 |
| 2,100,000 | 729,013.4 | 4,373,595 | 0.797 | −1.224 |
| 2,200,000 | 729,013.3 | 4,373,595 | 0.833 | −1.177 |
| 2,300,000 | 729,013.3 | 4,373,595 | 0.872 | −1.145 |
| 2,400,000 | 729,013.3 | 4,373,595 | 0.894 | −1.118 |
| 2,500,000 | 729,013.3 | 4,373,595 | 0.898 | −1.095 |
| 2,600,000 | 729,013.2 | 4,373,595 | 0.91 | −1.08 |
| 2,788,870 | 729,013.2 | 4,373,595 | 0.93 | −1.068 |

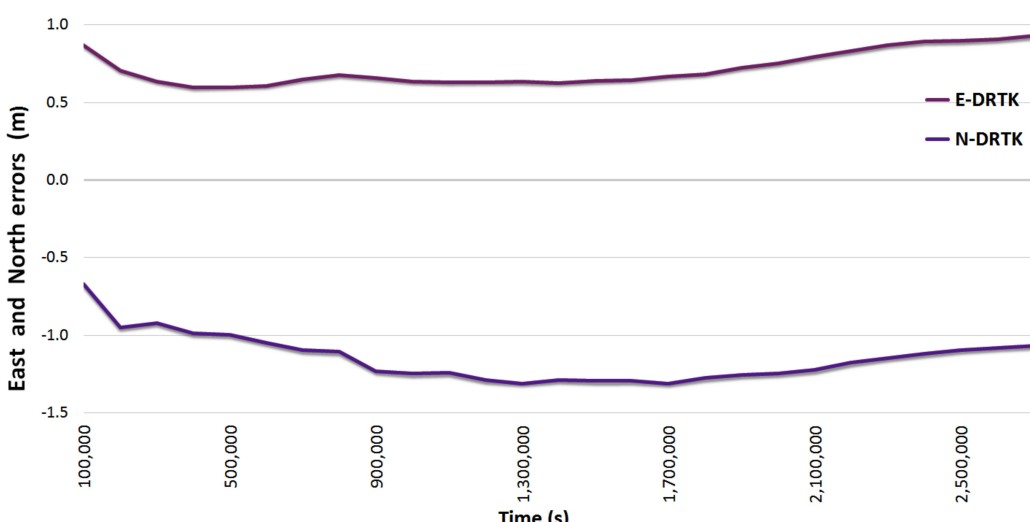

**Figure A3.** The relationship between the time and the East and North errors of Samsung Galaxy S3 device. East and North errors are the differences between the control point and coordinate means.

**Table A4.** Huawei Y330-U01 device period of recorded data, and coordinate means for E and N. Differences between the reference data and coordinate means (E-DRTK and N-DRTK). Real-time techniques were used to obtain the reference data: 729,008.787 m, 4,373,591.517 m.

| Period of Recorded Data (s) | E Mean (m) | N Mean (m) | E-DRTK (m) | N-DRTK (m) |
|---|---|---|---|---|
| 100,000 | 729,011 | 4,373,593 | −2.167 | −1.044 |
| 200,000 | 729,009.9 | 4,373,592 | −1.072 | −0.813 |
| 300,000 | 729,009.5 | 4,373,593 | −0.738 | −1.057 |
| 400,000 | 729,009.4 | 4,373,593 | −0.591 | −1.182 |
| 500,000 | 729,009.2 | 4,373,593 | −0.414 | −1.057 |
| 571,180 | 729,009.2 | 4,373,593 | −0.384 | −1.134 |
| 620,000 | 729,009.2 | 4,373,593 | −0.374 | −1.037 |
| 670,000 | 729,009.1 | 4,373,593 | −0.274 | −1.017 |
| 720,000 | 729,009.1 | 4,373,592 | −0.335 | −0.935 |
| 780,000 | 729,009.174 | 4,373,592.54 | −0.384 | −1.037 |

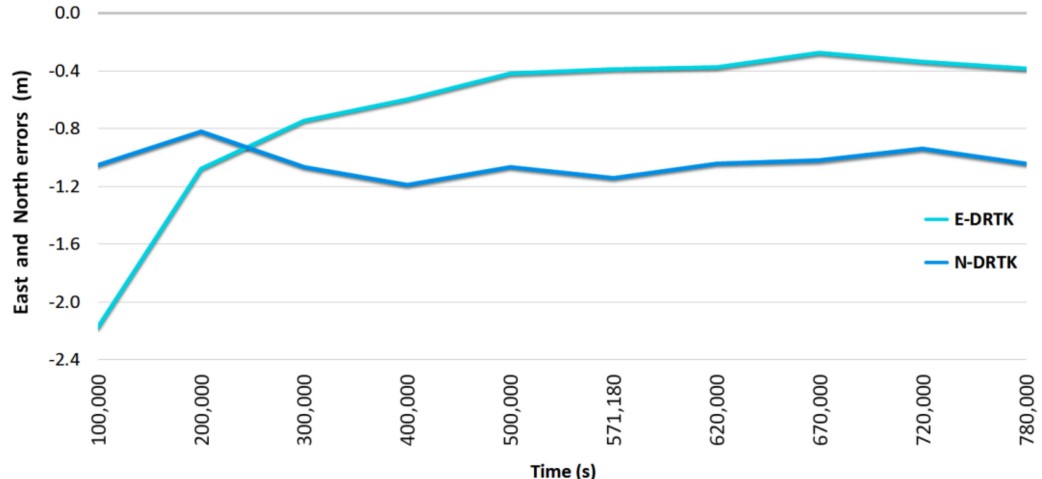

**Figure A4.** The relationship between the time and the East and North errors of Huawei Y330-U01 device. East and North errors are the differences between the control point and coordinate means.

**Table A5.** Xiaomi Mi 8 device period of recorded data, and coordinate means for E and N. Differences between the reference data and coordinate means (E-DRTK and N-DRTK). Real-time techniques were used to obtain the reference data: 729,008.787 m, 4,373,591.517 m.

| Period of Recorded Data (s) | E Mean (m) | N Mean (m) | E-DRTK (m) | N-DRTK (m) |
|---|---|---|---|---|
| 10,000 | 729,007.4 | 4,373,592 | 1.384 | −0.465 |
| 20,000 | 729,007.5 | 4,373,591 | 1.308 | 0.211 |
| 30,000 | 729,008 | 4,373,592 | 0.807 | −0.06 |
| 40,000 | 729,008.2 | 4,373,592 | 0.54 | −0.144 |
| 50,000 | 729,008.3 | 4,373,591 | 0.514 | 0.057 |
| 60,000 | 729,008.5 | 4,373,592 | 0.275 | −0.154 |
| 70,000 | 729,008.6 | 4,373,592 | 0.172 | −0.387 |
| 80,000 | 729,008.6 | 4,373,592 | 0.148 | −0.472 |
| 90,000 | 729,008.6 | 4,373,592 | 0.165 | −0.416 |
| 100,000 | 729,008.4 | 4,373,592 | 0.341 | −0.457 |
| 110,000 | 729,008.4 | 4,373,592 | 0.368 | −0.441 |
| 120,000 | 729,008.4 | 4,373,592 | 0.353 | −0.474 |
| 130,000 | 729,008.4 | 4,373,592 | 0.39 | −0.471 |
| 140,000 | 729,008.4 | 4,373,592 | 0.373 | −0.427 |
| 150,000 | 729,008.4 | 4,373,592 | 0.366 | −0.438 |
| 160,000 | 729,008.4 | 4,373,592 | 0.343 | −0.383 |
| 170,000 | 729,008.5 | 4,373,592 | 0.327 | −0.273 |
| 180,000 | 729,008.5 | 4,373,592 | 0.321 | −0.338 |
| 190,000 | 729,008.5 | 4,373,592 | 0.309 | −0.426 |
| 200,000 | 729,008.4 | 4,373,592 | 0.361 | −0.399 |
| 210,000 | 729,008.4 | 4,373,592 | 0.346 | −0.373 |
| 217,587 | 729,008.5 | 4,373,592 | 0.307 | −0.346 |

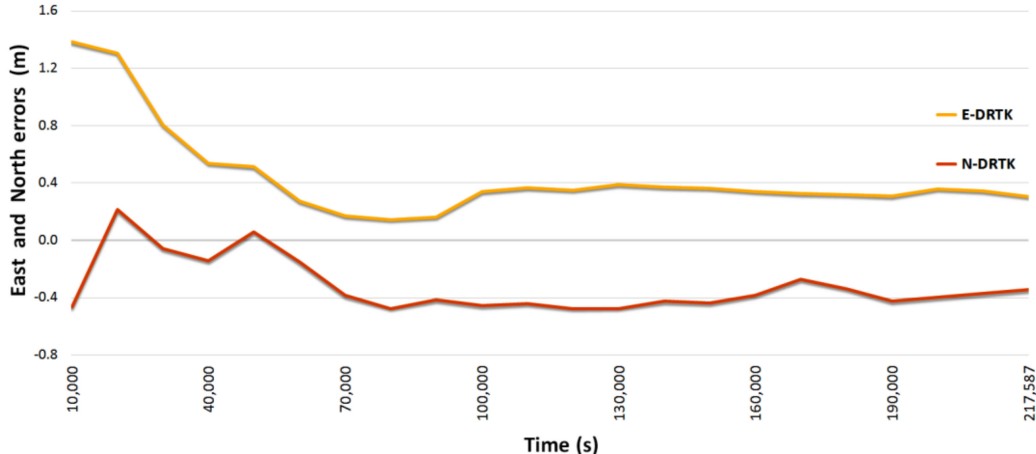

**Figure A5.** The relationship between the time and the East and North errors of Xiaomi Mi 8 device. East and North errors are the differences between the control point and coordinate means.

**Table A6.** The $d_E$, $d_N$ error from the E, N coordinates obtained from the mean of different numbers of observations.

| Observations | Vertex 1 (m) | Vertex 1 (m) | Vertex 2 (m) | Vertex 2 (m) | Vertex 3 (m) | Vertex 3 (m) |
|---|---|---|---|---|---|---|
| 500 | 0.86 | −1.04 | −0.25 | −0.18 | 1.92 | −2.85 |
| 1000 | 1 | −1 | −0.13 | 0.41 | 1.64 | −1.28 |
| 1500 | 0.66 | −0.46 | −0.09 | 0.53 | 1.5 | −0.57 |
| 2500 | 0.66 | −1.01 | 0.18 | 0.69 | 1.06 | 0 |
| 5000 | −0.48 | −1.36 | −0.18 | 0.38 | 0.57 | −0.91 |
| 10,000 | −0.18 | −0.33 | −0.71 | −0.56 | 0.29 | −1.25 |
| 15,000 | 0.12 | 0.25 | −0.79 | −0.63 | 0.41 | −0.62 |
| 19,542 | −0.08 | −0.11 | −0.5 | −0.7 | 0.61 | −0.14 |

**Table A7.** The $d_E$, $d_N$ error from the E, N coordinates obtained from the expression (27) number of observations.

| Observations | Vertex 1 (m) | Vertex 1 (m) | Vertex 2 (m) | Vertex 2 (m) | Vertex 3 (m) | Vertex 3 (m) |
|---|---|---|---|---|---|---|
| 500 | −0.84 | 1.35 | −0.84 | 1.35 | −0.84 | 1.35 |
| 1000 | −0.84 | 0.62 | −0.83 | 0.62 | −0.83 | 0.62 |
| 1500 | −0.69 | 0.17 | −0.69 | 0.17 | −0.69 | 0.17 |
| 2500 | −0.37 | 0.11 | −0.37 | 0.11 | −0.37 | 0.11 |
| 5000 | 0.03 | 0.63 | 0.03 | 0.63 | 0.03 | 0.63 |
| 10,000 | 0.2 | 0.71 | 0.2 | 0.71 | 0.2 | 0.72 |
| 15,000 | 0.08 | 0.33 | 0.08 | 0.33 | 0.08 | 0.33 |
| 19,542 | −0.01 | 0.32 | −0.01 | 0.32 | −0.01 | 0.32 |

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
