# Peer review of "An Approach to Improving GNSS Positioning Accuracy Using Several GNSS Devices"

_remotesensing, doi:10.3390/rs13061149_

Round 1
Reviewer 1 Report
The paper: 'An Approach to Improve GNSS Positioning Accuracy with Several GNSS Devices' is about an alternative method of processing GNSS data in SPP to increase its otherwise poor accuracy and not useful for precision surveying.
The paper is interesting and well presented, here are some of my observations to improve it further.
- You should be more detailed about the advantages you think you can achieve with this approach, for example when you say: "The advantages of working un-der SPP include not requiring carrier phase observations, external corrections or high-precision products." I would point out that this is important because only few mobile phones can write phase observations including the Xiaomi MI8 but, for example, inexplicably the Xiaomi Mi9 does not do so it only writes code information [1]. This consideration reinforces the purpose of your approach.
-Equation 1: I'm not sure it makes sense to write in a magazine like this how to calculate the variance, I personally would omit it. Or if you think it can't be removed, I would write something like: "...as is well known..."
- I would summarise a few passages from page 5 to page 8, the obvious ones can be omitted
-The sentence: "GNSS data was collected using our own app that logs the time, latitude, and longi-tude. The recorded data were converted into the Universal Transverse Mercator (UTM-ETRS89) projection. The European Terrestrial Reference System of 1989 (ETRS89) has been the official system in Spain since 2012." does not clarify exactly what you have done: have you simply projected the WGS84 Geographic coordinates into UTM using the equations of the representation or have you considered the different materialisation. When naming a reference system I usually add the EPSG code to make sure there are no misunderstandings.
- on page 12 when you talk about "receptors" do you mean receivers?
-in the first paragraph of the conclusions what exactly do you mean by "exactitude"?
- more generally, I think it would be useful to know what the possible applications of this approach might be, but it is interesting from a methodological point of view.
Best regards
[1]BAIOCCHI, V.; DEL PIZZO, S.; PUGLIANO, G.: ONORI, M., ROBUSTELLI, U., TROISI, S., VATORE, F. and LEÓN TRUJILLO, F.J., 2020. Use of the sensors of a latest generation mobile phone for the three-dimensional reconstruction of an archaeological monument: The survey of the Intihuatana stone in Machu Picchu (Peru'), IOP Conference Series: Materials Science and Engineering 2020.
[2]Aggrey, J.; Bisnath, S.; Naciri, N.; Shinghal, G.; Yang, S. Multi-GNSS precise point positioning with next-generation smartphone measurements. J. Spat. Sci. 2019, 65, 79–98.
[3]Hesselbarth, A., & Wanninger, L. (2020). Towards centimeter accurate positioning with smartphones. Paper presented at the 2020 European Navigation Conference, ENC 2020, doi:10.23919/ENC48637.2020.9317392
Reviewer 2 Report
The manuscript evaluates the reliability of observations from a large number of low-cost devices through data segmentation and bootstrapping statistical methods, and improves the positioning accuracy of the SPP by applying external constraints. This study is very interesting and meaningful, so in this sense the manuscript is suitable for publication in Remote Sensing. However, before publication the manuscript should be improved with respect to the following points:
- Precise Point Positioning technique and precise point position are both abbreviated as PPP, which makes me confused.
- The logic of the manuscript seems a little confused, and it would be nice to see improvements in this area.
- Page 8, I am confused that the estimated coordinate error can reach decimeter level, so what is the significance of the comparison between the final result and the estimated coordinate?
- Page 9, “The best results have been obtained by Garmin device followed by Xiaomi Mi 8. These two devices have operational capability of more GNSS systems”. Does that mean that all traceable systems are being utilized?
- In my understanding, the study in the manuscript is to enhance the strength of the model by adding geometric configuration constraints to obtain high-precision position solutions, is that right?
Round 2
Reviewer 2 Report
I agree with the arguments given in the rebuttal letter and feel quite satisfactory with the revised work, which I think can now be published as is.
Author Response
The authors are grateful to the Editor and anonymous reviewers for their valuable suggestions and constructive comments.